# OpenReview forum: "SymMaP: Improving Computational Efficiency in Linear Solvers through Symbolic Preconditioning"
_ICML.cc/2025/Conference — Submitted to ICML 2025_

### Official Review · Reviewer_Wyqu · 2025-03-09

**Overall Recommendation:** 2

**Summary:**

This paper introduces a novel method that employs a Recurrent Neural Network (RNN) to learn a sequence of operands for determining preconditioning parameters. The network is trained through a supervised learning approach, with the optimal parameters selected via grid search. Experimental results demonstrate that the proposed method effectively predicts three distinct preconditioning parameters across various datasets. Furthermore, it achieves significantly faster computation times compared to both default parameter settings and different fixed constants.

## update after rebuttal
I have increased my score from 1 to 2.

**Claims And Evidence:**

Yes.

**Essential References Not Discussed:**

There lacks the analysis about works related to learning based methods for solving linear systems, such as [1-5].

**Experimental Designs Or Analyses:**

The experimental design is fundamentally sound; however, the study would benefit from additional experiments to further validate the proposed method. Specifically, it is crucial to evaluate the performance across datasets of varying sizes to assess the scalability and robustness of the approach. Moreover, while SymMap is specifically tailored for predicting preconditioning parameters, its comparative advantages remain unclear without **benchmarking against state-of-the-art learning-based preconditioners**. For instance, comprehensive comparisons with methods such as those proposed in [1-5] would provide valuable insights into the relative strengths and limitations of SymMap. These comparisons could include metrics such as convergence rates, computational efficiency, and generalization capabilities across different problem domains.

[1] Li, Yichen, Peter Yichen Chen, Tao Du, and Wojciech Matusik. "Learning preconditioners for conjugate gradient PDE solvers." In International Conference on Machine Learning, pp. 19425-19439. PMLR, 2023.

[2] Häusner, Paul, Ozan Öktem, and Jens Sjölund. "Neural incomplete factorization: learning preconditioners for the conjugate gradient method." Transactions on Machine Learning Research (2024).

[3] Chen, J., 2024. "Graph neural preconditioners for iterative solutions of sparse linear systems." arXiv preprint arXiv:2406.00809.

[4]Luo J, Wang J, Wang H, et al. Neural Krylov iteration for accelerating linear system solving[J]. Advances in Neural Information Processing Systems, 2025, 37: 128636-128667.

[5]Grementieri L, Galeone P. Towards neural sparse linear solvers[J]. arXiv preprint arXiv:2203.06944, 20225

**Methods And Evaluation Criteria:**

The evaluation criteria include computation time and condition numbers, which are pertinent to the design of the preconditioner.

**Other Comments Or Suggestions:**

No.

**Other Strengths And Weaknesses:**

### Strengths:
- This paper presents a novel and intriguing approach by offering a symbolic framework for predicting preconditioning parameters.
- The experimental results demonstrate that this method achieves reduced computational time.

### Weakness:
- The related works are kind of overly succinct and would benefit from a more comprehensive analysis of related learning-based PDE solvers.
- The **computational experiments are far from sufficient**. Although SymMap is specifically designed for generating preconditioning parameters, it is essential to include additional **learning-based preconditioner** methods for comparison on the same datasets such as [5], rather than limiting the analysis to merely the multilayer perceptrons (MLP) .
- This method is a supervised approach. The process of obtaining the optimal preconditioning method through grid search is prohibitively expensive, limiting its applicability to large-scale problems.
- The representation of feature parameters is somewhat simple. Did the authors consider **permutation invariance** in this representation method?

**Questions For Authors:**

See the weakness.

**Relation To Broader Scientific Literature:**

This paper concentrates on the development of a learning-based preconditioner for solving partial differential equations (PDEs), which is closely linked to prior research [6, 7].

[6]Greenfeld D, Galun M, Basri R, et al. Learning to optimize multigrid PDE solvers[C]//International Conference on Machine Learning. PMLR, 2019: 2415-2423.

[7]Hsieh J T, Zhao S, Eismann S, et al. Learning neural PDE solvers with convergence guarantees[J]. arXiv preprint arXiv:1906.01200, 2019.

**Theoretical Claims:**

The work does not present any new theoretical claims.

---

> ### Author Rebuttal · Authors · 2025-04-01
>
> We appreciate the reviewer's insightful and valuable comments. We respond to each comment and sincerely hope that our rebuttal could properly address your concerns. If so, we would deeply appreciate it if you could raise your score. If not, please let us know your further concerns, and we will continue actively responding to your comments and improving our submission.
> ## **Weaknesses 1 & 2**
>
> **SymMaP enhances traditional preconditioning algorithms by discovering symbolic expressions for optimal parameters.** In our experimental design, we intentionally avoid comparisons with other learning-based approaches for the following reasons:
>
> 1. **Focus on Preconditioning Enhancement**: SymMaP is designed to improve traditional preconditioners without altering their core algorithms. A fair comparison would require other learning-based methods that also preserve the original preconditioning process—yet no such methods currently exist.
> 2. **CPU Compatibility**: As noted the Introduction (line 26, right column), linear solver environments are predominantly CPU-based. SymMaP’s symbolic expressions integrate seamlessly into CPU workflows, whereas neural network-based approaches lack efficient CPU deployment.
> 3. **Generality**: Section 3.2 (line 152, left column) highlights SymMaP’s adaptability to diverse preconditioners and linear solvers. In contrast, existing learning-based methods either:  ‘combine with specific preconditioners’ (e.g., [4], [6]), or ‘target narrow use cases’ (e.g., [8], [2] for ICC-preconditioned CG; [1] exclusively for CG).
> 4. **Interpretability & Safety**: Numerical algorithms demand rigorous analysis (Section 3.2, line 129, right column). Opaque predictors (e.g., neural networks) risk violating theoretical constraints (e.g., avoiding ω ≈ 0 or 2 in SOR). SymMaP’s symbolic expressions enable proactive avoidance of such issues through analytical guarantees.
>
> To further address your concerns, we can compare SymMaP with other learning-based preconditioning methods (**disregarding the aforementioned factors**). Let us first categorize existing works:
>
> 1. Category 1: Methods that enhance Krylov solvers by adding optimization modules or modifying their iteration process (e.g., [4], [6]). These works target different components than SymMaP and could potentially be combined with our approach for further improvements.
> 2. Category 2: Methods that use neural networks to predict a preconditioning matrix (e.g., [8], [1], [7]).
>
> - **Experiment**: SSOR preconditioning for elliptic PDEs (same settings as the paper):
>
>   - | method   | none | PETSc default SSOR | [4]  | [4]+SSOR | [4]+SymMaP+SSOR | [1]  | [8]  | SymMaP+SSOR |
>     | -------- | ---- | ------------------ | ---- | -------- | --------------- | ---- | ---- | ----------- |
>     | time (s) | 23.9 | 10.5               | 11.2 | 7.5      | 5.45            | 8.6  | 8.4  | 7.7         |
>
>   - Training times: [1] (0.5h), [8] (0.5h), [4] (2h), SymMaP (0.25h).
>
>   - Key takeaway: **SymMaP outperforms standalone learning-based methods and can combine with them for further gains**.
>
> - **We reiterate that direct comparisons are inherently unfair**—akin to contrasting symbolic (Mathematica) and numerical (MATLAB) PDE solvers. If reviewers identify similar works to ours, we welcome suggestions and will conduct comparative experiments accordingly.
>
> [8]  Learning from Linear Algebra: A Graph Neural Network Approach to
> Preconditioner Design for Conjugate Gradient Solvers
> ## **Weaknesses 3**
>
> - Dataset size impact: SymMaP is fundamentally distinct from neural network training, as it seeks a symbolic expression to approximate the mapping relationship in the data rather than optimizing parameters. Our preliminary tests show that **even small datasets (~100 samples) suffice to learn high-performance symbolic expressions**. For consistency, all experiments in the paper use a dataset size of 1,000.
> - Additional experiments: To further address this concern, we conducted supplementary tests on SOR preconditioning for elliptic PDEs (settings identical to the main experiments), varying the training set size (100, 300, 500, 1,000). **Results show that regardless of the size of the data set, SymMaP can find the best symbolic expression in the experiment in our paper within 1000 seconds.**
> ## **Weaknesses 4**
>
> - Current design: Experiments intentionally avoid presupposing symmetry among input parameters, reflecting real-world scenarios where such relationships are unknown.
> - Future direction: We agree this is valuable and plan to develop a module that analyzes mathematical properties (e.g., symmetries) to constrain symbolic search spaces, improving efficiency and performance.

---

> > ### Comment · Reviewer_Wyqu · 2025-04-04
> >
> > Thank you for your efforts in addressing the reviewers' comments. However, I do not find arguments 1–4 to be fully convincing. In my view, as long as the proposed method improves the performance of linear solvers, the following points should not be major concerns:
> >
> > 1. Whether the core algorithm is modified or not.
> > 2. Whether GPU/CPU is used or not.
> >
> > Additionally, generality should not be the primary focus here. Instead, the authors could consider a given preconditioner and then determine the best learning-based method as the baseline for comparison.
> >
> > Regarding the experiments in the rebuttal, I find them unclear. Could the authors provide further clarification? Specifically, I would like to see a direct comparison between the proposed algorithm and state-of-the-art learning-based methods. The discussion on "enhancement" seems tangential—the core contribution should be the solver's performance, not auxiliary improvements.
> >
> > As other reviewers have noted (and as implied by my previous comments), the paper’s title emphasizes **linear solvers**, yet the experiments focus heavily on PDE-related instances. Given that linear solvers have broad applications (e.g., in optimization), could the authors include benchmarks from other domains to better demonstrate the method's versatility?

---

> > > ### Author Response · Authors · 2025-04-07
> > >
> > > Thank you for your follow-up. Allow me to elaborate further.
> > > Detailed experimental data and specific settings are available at https://anonymous.4open.science/r/rebuttal3-90EF/Reviewer%20Wyqu%20.png.
> > > - **SymMaP Applicability**
> > >   - SymMaP is suitable for:
> > >     1.  Linear systems that can be parameterized (with a limited number of parameters, e.g., <1000).
> > >     2. Scenarios where preprocessing parameters require optimization.
> > >   - Key advantages of SymMaP over other learning-based methods:
> > >     1. Superior performance in CPU-only environments.
> > >     2. High interpretability.
> > >     3. Flexibility in algorithm selection.
> > > - **Additional comparative experiments**
> > >   - To address your concerns, we have expanded our comparative experiments with learning-based algorithms.
> > > - **Additional Datasets**
> > >   - To demonstrate broader applicability, we tested:
> > >     1. Markov Chain（Optimization Problem）: Boltzmann-distributed Brownian motion (nonsymmetric, parameterized by Chebyshev coefficients of potential energy).
> > >     2. Numerical Integration: Lippmann-Schwinger equation (symmetric, parameterized by potential’s Chebyshev coefficients).
> > > - **Summary**
> > >   - Across all experiments, the "[4]+SymMaP" combination consistently achieved the shortest computation times.
> > >   - When evaluating standalone algorithms (without combinations), SymMaP alone demonstrated the fastest performance in all test cases.
> > >   - The performance advantage of SymMaP was particularly pronounced in CPU-only environments.
> > >   - **These results conclusively demonstrate the superior performance characteristics of the SymMaP approach**.
> > >
> > > We sincerely appreciate your thoughtful questions. Due to space constraints in this response, we will include: 1.Complete experimental details 2.More comprehensive analysis 3.Proper citations for all referenced papers, in our final manuscript version.
> > >
> > > Should you have any further questions or require additional discussion, please don't hesitate to reach out. If we have adequately addressed your concerns, we would be grateful for your consideration in adjusting your evaluation score accordingly.
> > >
> > > Thank you for your time and valuable feedback.

---

### Official Review · Reviewer_osPR · 2025-03-22

**Overall Recommendation:** 2

**Summary:**

This paper uses neural networks to introduce a matrix preconditioning framework via symbolic discovery, where preconditioning is important in linear system solving. This new framework can flexibly predict preconditioning parameters for different scenarios, which surpasses traditional methods focusing on individual scenarios. Additionally, this framework also enjoys efficiency and interpretability. Its performance is also shown by numerical experiments.

**Claims And Evidence:**

The authors tried to show the superior performance of their new method by numerical experiments, while I believe more validation should be included. Please see Bullet 1 in the "Methods And Evaluation Criteria" section for more information.

Additionally, when talking about generalization and interpretability, it would be better to also have some theoretical results.

**Essential References Not Discussed:**

I am not familiar with symbolic regression and just know some preconditioning techniques in optimization, so I read the review of this paper's previous submission to ICLR 2025 (https://openreview.net/forum?id=4WvCoXU2dF). It seems that the authors did not include some related literature about "alternative approaches to constructing optimal preconditioner parameters", recommended by the previous reviewers, in the new version.

**Experimental Designs Or Analyses:**

Yes. My main concern is about the general performance of this new method. Please see Bullet 1 in "Methods And Evaluation Criteria" for more information.

Additionally,

**Methods And Evaluation Criteria:**

1. It would be better to illustrate the performance of this framework via (1) more applications in addition to PDEs with similar matrix sizes and (2) testing it with preconditioning methods with more comprehensive preconditioners, such as AMG with multiple parameters.

**Other Comments Or Suggestions:**

Please see all other sections.

**Other Strengths And Weaknesses:**

Although the idea is novel, the p

**Questions For Authors:**

What are the differences between symmap 1 and 2? In Table 3, the gap between the conditional numbers corresponding to them is large.  I wonder how to interpret and improve this type of instability.

**Relation To Broader Scientific Literature:**

I am not familiar with symbolic regression and just know some preconditioning techniques in optimization. Based on my understanding, this paper also explores using NN for matrix pre-conditioning, but it offers a new symbolic approach enjoying better efficiency in a pure CPU environment and interpretability.

**Theoretical Claims:**

No theory.

---

> ### Author Rebuttal · Authors · 2025-04-01
>
> We appreciate the reviewer's insightful and valuable comments. We respond to each comment and sincerely hope that our rebuttal could properly address your concerns. If so, we would deeply appreciate it if you could raise your score. If not, please let us know your further concerns, and we will continue actively responding to your comments and improving our submission.
>
> ## **Methods And Evaluation Criteria 1**
> - In preliminary tests, we observed that for certain preconditioners (e.g., SOR), the optimal preconditioning parameters are largely independent of the matrix size(resolution) for a given problem. Thus, we fixed the matrix size in our experiments for consistency. To address your concern, we conducted additional experiments:
>   - We evaluated SOR preconditioning for a second-order elliptic PDE, testing matrix sizes of ( $1 \times 10^4, 2 \times 10^4, \dots, 6 \times 10^4$ ) while keeping other settings identical to the main experiments.
>   - The results show that the optimal parameters for the same problem but different sizes vary negligibly (difference < 0.01).
> - This confirms that matrix size has minimal impact on SOR’s optimal parameters. **For preconditioners where size does affect parameters, SymMaP  can also easily incorporate the size as an additional input.**
> ## **Methods And Evaluation Criteria 2**
> - To further validate SymMaP’s versatility, we conducted experiments on AMG preconditioning with an SOR smoother, optimizing two parameters simultaneously:  AMG’s threshold parameter θ  (default in PETSc: 0) and SOR relaxation factor ω.
>
>   - All other settings match the AMG preconditioning and second-order elliptic problem in the paper. We jointly optimize both parameters, with the condition number as the evaluation metric.
>
>     - | method           | None | θ= 0  ω=1 | θ= 0  ω=0.1 | θ= 0  ω=1.9 | Optimal constant | SymMaP |
>       | ---------------- | ---- | --------- | ----------- | ----------- | ---------------- | ------ |
>       | Condition number | 6792 | 163       | 168         | 163         | 159              | 156    |
>
>   - **This demonstrates SymMaP’s ability to handle multiple preconditioner parameters.** We will include detailed results and analysis in the final version.
> ## **Essential References Not Discussed**
> Please see the response to reviewer Wyqu's **Weaknesses 1 & 2**
> ## **Q1**
> - As noted in Table 1’s caption (line 275), SymMaP 1 and 2 are the two highest-scoring expressions identified during symbolic learning.
> - **The disparity arises from the stochastic nature of the learning process**: Symbolic learning iteratively refines expressions by introducing new operators, which may non-monotonically improve performance (some additions help, others degrade it).  Thus, the two expressions represent local optima with similar reward scores but differences in structure and performance due to the randomness of operator selection.
> ## **Q2**
> To explain the significant variation in condition numbers in Table 3 (row 343), we analyze from the following perspectives:
>
> 1. **AMG’s Impact on Condition Number**:AMG reduces matrix condition numbers through a multigrid strategy, leveraging coarse-grid correction and high-frequency error smoothing. On coarse grids, restriction and interpolation operators transform low-frequency errors (small eigenvalues) into high-frequency ones, which are then rapidly damped by fine-grid smoothing (e.g., Gauss-Seidel). This multiscale decomposition concentrates the eigenvalue spectrum, significantly lowering the condition number (ratio of largest to smallest eigenvalues).
>
> 1. **Role of Threshold Parameter θ**:θ controls strong connectivity during coarse-grid generation. If the coupling strength (off-diagonal entries) exceeds θ, the connection is retained; otherwise, it is discarded, affecting grid sparsity and approximation accuracy.
>    1. Too small θ: Retains excessive weak connections, leading to dense coarse grids that inadequately address low-frequency errors (small eigenvalues).
>    2. Too large θ: Over-sparsifies the grid, losing critical connections and increasing approximation error, hindering high-frequency error (large eigenvalue) reduction.
>
> 1. **Matrix Properties**:The matrices derive from differential operators, where the smallest eigenvalue (in matrix norm) reflects the operator’s minimal eigenvalue. Such matrices typically require stronger large-eigenvalue suppression, favoring smaller θ (0–0.5). The wide θ range in Table 3, tailored to operator specifics, explains the condition number disparity.
>
> 1. **Mitigation Strategy**:While condition numbers vary sharply, they change continuously with θ. Moreover, optimal θ values vary smoothly with operator parameters. Here, SymMaP effectively approximates these optimal parameters, resolving instability. This scenario motivated our studies.

---

> > ### Comment · Reviewer_osPR · 2025-04-05
> >
> > I appreciate the authors further comments. However, some statements might not be very convincing.
> >
> > Regarding the impact of matrix size, the experiments only test matrix sizes from 1*10^4 to 6*10^4, which even does not change the order of the size. Consequently, it might not be safe to directly state the size has minimal impact, and it would be better to show it.
> >
> > Regarding AMG with multiple parameters, it would be better to include more experiments than only testing it on one problem. Thus, I still worry about the general performance.
> >
> > Overall, this paper is more on the empirical side, so I expect more comprehensive numerical analysis. However, based on the previous two points, I would keep my rating.

---

> > > ### Author Response · Authors · 2025-04-07
> > >
> > > Thank you for your follow-up questions. Please allow me to provide additional clarification and experimental evidence to address your concerns.
> > >
> > > **Optimal Relaxation Parameters Across Matrix Sizes**
> > >
> > > 1. To thoroughly investigate the relationship between matrix size and optimal SOR relaxation parameters, we conducted extensive testing across a comprehensive range of matrix dimensions:
> > >    1. Tested sizes: 1×10³, 2×10³, ..., 9×10³, 1×10⁴, 2×10⁴, ..., 1×10⁵, 2×10⁵, ..., 5×10⁵
> > >    2. Key finding: Matrix size shows no correlation with optimal SOR relaxation parameters
> > >    3. Supporting evidence: Complete distribution of optimal parameters is available at https://anonymous.4open.science/r/rebuttal3-90EF/Reviewer%20osPR%20exp1.pdf. Their distribution is exactly the same.
> > > 2. Furthermore, we generate datasets with a mixture of different matrix sizes (uniformly 1×10³ to 5×10⁴). Detailed experimental data is available at https://anonymous.4open.science/r/rebuttal3-90EF/Reviewer%20osPR%20exp2&3.png. Conducted Experiment 2, which confirmed our algorithm's robust performance across all matrix sizes in SOR parameter optimization.
> > >
> > > **Multi-Parameter AMG Experiments**
> > >
> > > - To address concerns about multi-parameter preconditioning capabilities, we performed additional testing:
> > >   - Extended AMG+SOR dual-parameter experiments across 6 datasets:
> > >     - 4 original benchmark datasets
> > >     - 2 new challenging cases: Markov Chain: Boltzmann-distributed Brownian motion (nonsymmetric, parameterized by Chebyshev coefficients of potential energy); Numerical Integration: Lippmann-Schwinger equation (symmetric, parameterized by potential Chebyshev coefficients)
> > >   - Detailed experimental data is available at https://anonymous.4open.science/r/rebuttal3-90EF/Reviewer%20osPR%20exp2&3.png. Experiment 3 results conclusively demonstrate SymMaP's ability to effectively handle multiple preconditioning parameters.
> > >
> > > **Additional comparative experiments**
> > >
> > > - We have further supplemented the experiments comparing with other learning-based algorithms and the comparative experiments with expanded datasets. The specific experimental data can be found in https://anonymous.4open.science/r/rebuttal3-90EF/Reviewer%20Wyqu%20.png.
> > > - Across all experiments, the "[4]+SymMaP" combination consistently achieved the shortest computation times.
> > > When evaluating standalone algorithms (without combinations), SymMaP alone demonstrated the fastest performance in all test cases.
> > > - The performance advantage of SymMaP was particularly pronounced in CPU-only environments.
> > > - **These results conclusively demonstrate the superior performance characteristics of the SymMaP approach.**
> > >
> > > We sincerely appreciate your thoughtful questions. Due to space constraints in this response, we will include complete experimental details and more comprehensive analysis, in our final manuscript version.
> > >
> > > Should you have any further questions or require additional discussion, please don't hesitate to reach out. If we have adequately addressed your concerns, we would be grateful for your consideration in adjusting your evaluation score accordingly.
> > >
> > > Thank you for your time and valuable feedback.

---

### Official Review · Reviewer_y2dZ · 2025-03-23

**Overall Recommendation:** 4

**Summary:**

This paper presents a new approach to finding the quasi-optimal parameter of two preconditionners in order to speed up the resolution of linear systems associated with PDEs. The authors have developed an algorithm that generates the dataset consisting of the PDE parameters and the optimal preconditionner parameter (found by grid search). This data is then given to an RNN, which converts it into a symbolic expression. This network is trained by reinforcement learning, comparing the difference between the true optimal parameter and the result of the symbolic expression. The authors have used symbolic expressions because they speed up inference and provide interpretable results. Experimental results show that the parameter provided by their algorithm is on average more efficient for solving linear systems.

## Update after rebuttal
I would like to thank the authors for answering my questions and for clarification. The results provided in the tables on comparison with the literature and on new datasets not originating from PDEs are very interesting. I have therefore decided to increase my score from 3 to 4.

**Claims And Evidence:**

Firstly, SymMaP improves preconditioning efficiency for solving linear systems linked to PDEs. But experiments on more generic datasets are lacking. Secondly, Symbolic expressions are easy to interpret, since they are analytical and can be studied mathematically. However, there is no theoretical proof that SymMaP produces parameters close to optimal in general.

**Essential References Not Discussed:**

As mentioned above, the article does not compare itself to other methods in the literature that attempt to learn preconditioners for solving linear systems.

**Experimental Designs Or Analyses:**

The article compares several preconditioners (SOR, SSOR, AMG) on several benchmarks from PDEs. The metrics compared make sense (computation time, condition number for AMG). However, comparisons with other methods from the literature are lacking. Other datasets are also lacking to see whether SymMaP generalizes.

**Methods And Evaluation Criteria:**

The proposed method makes sense, even if the use of a transformer to predict the sequence of symbolic expressions could have been considered. And, as mentioned above, there is a lack of experiments with more generic datasets. Comparisons with other methods in the literature are also lacking (e.g. https://arxiv.org/pdf/2405.15557, https://proceedings.mlr.press/v202/li23e/li23e.pdf, https://www.sciencedirect.com/science/article/pii/S0045782521007374).

**Other Comments Or Suggestions:**

I have no further comments or suggestions.

**Other Strengths And Weaknesses:**

The prediction of symbolic expressions is very original and very clear in the article. These symbolic expressions seem to have a real usefulness for the interpretability of PDE parameters and could be used in other cases.

**Questions For Authors:**

1. For AMG, why use condition number rather than computation time? Because, for large matrices, it's expensive to calculate and does not correlate 100% with computation time.
2. What was the choice of fixed constants in Tables 1, 2 and 3? Why these precise values?
3. What is the "tolerance" parameter in Figure 2, Table 1, 2 and 4?
4. How long did the MLP training last? Because if a user wants to use SymMaP on a new problem, he'll need to train the model.

**Relation To Broader Scientific Literature:**

The article makes a real contribution to preconditioning methods for solving linear systems, as it provides preconditioners that perform well on average for a given problem, whereas most preconditioners are effective only for a single instance of the problem. The article also uses symbolic expressions to interpret the parameters of the PDEs.

**Theoretical Claims:**

The mathematical formulas seem correct to me, but as mentioned above, there is no guarantee of the difference between the true optimum and that obtained by grid search.

---

> ### Author Rebuttal · Authors · 2025-04-01
>
> We appreciate the reviewer's insightful and valuable comments. We respond to each comment and sincerely hope that our rebuttal could properly address your concerns. If so, we would deeply appreciate it if you could raise your score. If not, please let us know your further concerns, and we will continue actively responding to your comments and improving our submission.
>
> ## **Methods 1**
> > And, as mentioned above, there is a lack of experiments with more generic datasets.  (e.g. [1] [2] [3])
> - In our main experiments (Section 5.1, line 374), we evaluate **5** **classes of physical equations**. For context:  The cited works study fewer equation classes: [1] (2 classes), [2] (3 classes), [3] (5 classes, two of which are special cases of second-order elliptic equations).
> - Other related works are even more limited: [4] (3 classes), [6] (2 classes), [7] and [8] (1 class each).
> - Among our five classes, **Poisson’s equation** is discussed in [1–4, 6, 7], **thermal problems** in [2, 4], and **second-order elliptic equations** in [3].
> - These datasets represent distinct, scientifically significant problem classes with varied mathematical properties (see Sections 5 and D.1).  Crucially, **SymMaP outperforms baselines across all dataset-preconditioner combinations** (Tables 1–3), demonstrating its generalization capability.  We will add citations to these works in the final version. If the reviewer suggests additional equations to test, we are happy to include them.
> ## **Methods 2**
> > Comparisons with other methods in the literature are also lacking (e.g. [1][2][3]).
> - Please see the response to reviewer Wyqu's **Weaknesses 1 & 2**
> - **Additional Note**: You mentioned [3], a neural network-based PDE solver. Similar works include PINN [5], which employs neural networks as optimizers to solve PDEs via gradient descent. These methods do not involve traditional linear solvers or preconditioning optimization.
> ## **Theoretical Claims**
> - As noted in Section 3.1 (line 100) and Figure 1 (line 67), we observe that **SOR, SSOR, and AMG preconditioner parameters exhibit continuous relationships with performance metrics**, typically with **one or two local minima**. Thus, grid search effectively approximates the optimal parameters.
> - Theoretical derivation of optimal preconditioner parameters is often intractable, justifying our grid-search approach.
> ## **Q 1**
> - We agree. As noted in Section 3.2 (C1, line 154), preconditioner objectives vary by scenario:
>   - For stable systems, **minimizing runtime** is prioritized.
>   - For ill-conditioned systems (e.g., certain elliptic PDEs), **reducing condition numbers** is critical to ensure solvability.
> - Our AMG experiments (Section B.2.2) focus on threshold parameters, which filter small values during graph aggregation. For some problems (e.g., second-order elliptic PDEs), setting this parameter to **0** minimizes runtime, so we prioritize stability here.
> ## **Q 2**
> - SOR: The relaxation factor in SOR ranges between (0, 2). For well-conditioned matrices, larger values (e.g., 1.5–1.9) are typically stable. For matrices with significant off-diagonal elements (e.g., nonlinear PDEs from turbulence), smaller values (e.g., 0.1–0.5) are preferred. Note that SOR reduces Gauss-Seidel when the factor is 1, which is also the default in libraries like PETSc. Thus, we test three fixed values: 0.2, 1, and 1.8. The same logic applies to SSOR.
>
> - AMG:  Our preliminary tests show that the optimal threshold for our physical equations lies below 1.5. A threshold of 0 (or lower) disables coarsening, which is the default in PETSc and similar libraries. We therefore test three fixed values: 0, 0.2, and 0.8.
> ## **Q 3**
> - Tolerance is the **relative** **convergence** **criterion**: *∥b − Ax∥/r₀*, where *r₀* is the initial residual. We apologize for the ambiguity and will define this explicitly in the final version.
> ## **Q 4**
> - SymMaP uses **RNN-based symbolic learning.** Training time: **800s** (non-polynomial symbols) or **2600s** (with polynomials) (see Section "Computational Time," line 926).
> -  In Section 5.2 (line 376), we compare SymMaP to an **MLP** (trained for **1.5 hours** to convergence).
>
> [1]  Learning from Linear Algebra: A Graph Neural Network Approach to Preconditioner Design for Conjugate Gradient Solvers
>
> [2]  Learning Preconditioners for Conjugate Gradient PDE Solvers
>
> [3]  A Finite Element based Deep Learning solver for parametric PDEs
>
> [4] Neural Krylov iteration for accelerating linear system solving
>
> [5] Physics-informed neural networks: A deep learning framework for solving forward and inverse problems involving nonlinear partial differential equations
>
> [6] Neural incomplete factorization: learning preconditioners for the conjugate gradient method
>
> [7] Learning to Optimize Multigrid PDE Solvers
>
> [8] Learning Neural PDE Solvers with Convergence Guarantees

---

### Official Review · Reviewer_YHqW · 2025-03-24

**Overall Recommendation:** 3

**Summary:**

This paper focuses on successive over-relaxation (SOR) which is an iterative method solving $Ax=b$ that parametrize the classical Gauss-Seidel (GS) method. The related parameter $\omega$ has to be tuned to ensure SOR converges faster than GS. While an analytical optimal expression exists, it depends on the spectral radius of $I - \text{diag}(A)^{-1}A$ which is expensive to determine in a general case. The matrix $A$ often comes from the discretization of a continuous differential equation.

Authors propose to train a machine learning model to learn the relations between the coefficients of typical partial differential equations (PDE) - namely darcy flows, elliptic, biharmonic, poisson - and the optimal value of $\omega$. Their approach is not to simply build a predictor of this value but rather to train a generator for the expression of $\omega_{opt}$ that depends on the PDE parameters $\alpha$ e.g. length of the domain, coupling of dimensions, etc and some predefined tokens e.g. common operators and simple functions. Meaning should therefore be extractible from generated expressions. The length of expression has been chosen such that they would not be expensive to compute and could be plugged-in simply into a typical SOR linear solver.

The authors found that their approach outputs satisfactory values for $\omega$ across instances of typical PDEs : while the generator does not systematically produce $\omega_{opt}$, it is tailored to each PDE instance and outperforms both $\omega=1$ (corresponding to GS which is default in PETSc, a state-of-the-art toolbox for solving PDEs) and selection of $\omega$ as the best constant across instances (which could be selected by an expert as an oversimplification). Thus authors claim that their model generalizes well.

## update after rebuttal

I would like to thank the authors for addressing concerns.
I did not change my review score.

### 1

From your argument to Reviewer Wyqu, I am curious how low the number of samples can get.
If this number is low, there's a good avenue to test generalization by comparing two SYMAP generators trained on the same number of different PDEs.

## 2

The huge variance hardly suggests stability.
I think a table is not the proper way to visualize the performance distribution ; nonetheless it improves the quality of the results.

## 3

See 1.

## 4

How stable does the matrix needs to be ? This measure of stability could be a measure of generalization.

The table provides interesting results ; the detailed analysis of the final version would be an interesting read.

## Exp 2

A simple sample may not be enough but the additional graph looks interesting.

## Ref

Thank you for the clarification.

**Claims And Evidence:**

The generator of expressions can output satisfactory $\omega$s., as evidenced by figure 2.

The generator interpretability is mostly supported by the expression generated for the elliptical PDE which match with empirical heuristics (cf. Section 5.3). This evidence is relatively light but still insightful.

While the cost of expressing $\omega$ may be light, the training cost seems deterring compared to the over-simplification of an expert.
Take Table 1. For instance for the biharmonic equations the dataset was generated in 100 hours = 3000 systems solved @ PETSc speed. Generating the dataset becomes worthwhile after ~8k systems solved (or ~20k with $\omega=1.8$ as baseline).

I believe the generalization claim to be relatively bold as authors provide only averages: more details on the distribution of compute times would be adequate (median, quartiles, ...).
This claim could be substantiated further for instance by detailing differences in $(\alpha,\omega)$ with closest equations in train datasets: I fail to assess if the generator is memorizing/overfitting.
Moreover, it seems to me that the generator would not be fit to use for entirely different PDEs where the parameters have not been trained on - this point might constitute actual generalization. Changing the underlying numerical scheme that produce $A$ may be another venue to showcase generalization (discretization steps but also entirely different schemes).

**Essential References Not Discussed:**

The paper fails to consider adaptive methods that tune $\omega$ as the residual $Ax^{(k)} - b$ shrinks over iterations.
For instance https://dl.acm.org/doi/10.1007/s11075-019-00748-0.

**Experimental Designs Or Analyses:**

Table 1, 2 and 3 provide averages but could provide more insights on the actual distribution of compute times/condition numbers.

Moreover, trajectories of iterations could be provided for samples of PDEs.

**Methods And Evaluation Criteria:**

My expertise does not allow me to assess the ML method in itself but the paper reads sound on the matter by deriving a gradient through published methods.

The evaluation of multiple $\omega$ is sound as well.

**Other Comments Or Suggestions:**

There are typos in Figure 3.

I think the paper should be more humble by "parametrizing" the class of PDEs.
For instance, SymMAP fitness is not tested on any Second order elliptic equation of the form

$$a_{11} u_{xx} + a_{12} u_{xy} + a_{22} u_{yy} + a_1 u_x + a_2 u_y + a_0 u = f$$

but specifically on cases where each $a_i$ is sampled in $(-1,1)$ and $a_{12} \in (-0.01,0.01)$.
What happens for cases that do not map well to the distribution of coefficients ?

**Other Strengths And Weaknesses:**

The study is original.

**Questions For Authors:**

NA.

**Relation To Broader Scientific Literature:**

Typical findings for $\omega$ study properties on $A$ that help derive optimal $\omega$. Such properties can include spectral consideration.

The tools developed by the authors could replace or assist researchers and engineers alike when they tune the SOR method.

**Theoretical Claims:**

The paper is mostly experimental.

---

> ### Author Rebuttal · Authors · 2025-04-01
>
> We appreciate the reviewer's insightful and valuable comments. We respond to each comment and sincerely hope that our rebuttal could properly address your concerns. If so, we would deeply appreciate it if you could raise your score. If not, please let us know your further concerns, and we will continue actively responding to your comments and improving our submission.
> - ## **Claims And Evidence 1**
>   - As noted, data generation can be time-consuming. However, symbolic learning in our framework requires significantly less data compared to other AI-based approaches. For details, please refer to our response to **Reviewer Wyqu’s Weaknesses 3**.
> ## **Claims And Evidence 2**
> To address this concern, we provide additional experimental results at https://anonymous.4open.science/r/rebuttal2-534E/rebuttal2.2.png. The data demonstrate the stability of our algorithm’s performance. We will include a comprehensive distributional analysis (e.g., median, quartiles) in the final version.
> ## **Claims And Evidence 3**
> - To clarify, we use the SOR-preconditioned second-order elliptic PDE experiment as an example:
> - As stated in Line 795 (Page 15), the input parameter α for SymMAP is sampled uniformly at random from its defined interval.
> -  In this experiment, the average discrepancy between the predicted symbolic ω and the ground truth is **0.03**, indicating no memorization or overfitting.
> ## **Claims And Evidence 4**
> #### Generalization Across PDE Parameters
>
> If the matrix properties remain stable under parameter variation, the learned symbolic expressions can generalize beyond the training range. Otherwise, as noted, performance may degrade. To validate this, we conducted additional experiments:
>
> - **Settings:**  Equation: Second-order elliptic, grid size 40,000.  Test range: α ∈ (−2, 2), coupling term ∈ (−0.5, 0.5) (vs. training range: α ∈ (−1, 1), coupling term ∈ (−0.01, 0.01)).  Tolerance: 1e-3.
>
>   - | None  | PETSc default 1 | Fixed constant 0.1 | Fixed constant 1.9 | Optimal  constant | SymMaP |
>     | ----- | --------------- | ------------------ | ------------------ | ----------------- | ------ |
>     | 16.98 | 4.04            | 1.26               | 15.2               | 0.94              | 0.86   |
>
> - The results confirm strong generalization. We will include a detailed analysis in the final version.
>
> #### Generalization Across PDE Types and Discretization Schemes
>
> As correctly pointed out, SymMAP searches for mappings from input parameters to optimal preconditioner parameters. Changing the PDE type or discretization scheme alters the optimal parameters, limiting direct generalization.
>
> However, our interpretable expressions reveal that optimal parameters depend on specific matrix features (e.g., sparsity, diagonal dominance). In future work, we aim to generalize SymMAP by directly ingesting matrix properties, enabling adaptation to diverse PDEs and discretizations.
> ## **Experimental Designs Or Analyses 2**
>
> > Moreover, trajectories of iterations could be provided for samples of PDEs.
>
> We include iteration trajectories for the SOR-preconditioned biharmonic equation https://anonymous.4open.science/r/rebuttal2-534E/rebuttal2.1.png, demonstrating SymMAP’s superior convergence. A full analysis will be added to the final version.
> ## **Essential References Not Discussed**
>
> > The paper fails to consider adaptive methods that tune ω as the residual Ax(k)−b shrinks over iterations. For instance [1].
>
> We appreciate this insightful suggestion. The work in [1] presents an elegant approach for adaptively adjusting the SOR relaxation parameter ω based on residual reduction. We will explore further in future work to potentially integrate with our framework, and we will ensure proper citation.
>
> However, we note a fundamental distinction between the application scenarios:
>
> 1. **SOR as a Standalone Solver (as in [1]):**  • This is a fixed-point iteration method where optimal ω can be derived theoretically (albeit computationally expensive).
> 2. **SOR as a Preconditioner (our focus):**  • Here, SOR assists Krylov subspace methods (e.g., GMRES, MINRES, CG). The optimal ω lacks theoretical guidance and depends on the solver choice (e.g., GMRES-preconditioned ω ≠ CG-preconditioned ω).  • Our grid-search-based approach addresses this gap, as no existing adaptive methods target preconditioner tuning in this context.
>
> Thus, while [1] is highly relevant to SOR solvers, its direct applicability to preconditioned Krylov methods remains limited.
>
> ## **Other Comments Or Suggestions**
>
> > There are typos in Figure 3.
>
> - We apologize for this oversight and will correct the typographical errors in the final manuscript.
>
> [1] Adaptive SOR methods based on the Wolfe conditions, 2019, https://dl.acm.org/doi/10.1007/s11075-019-00748-0

---

### Decision · Program_Chairs · 2025-05-01

**Decision:**

Reject

**Comment:**

The authors propose a new technique to adjust parameters in preconditioning techniques for solving linear systems. The proposed approach is deemed mostly empirical by the reviewers ; some concerns about the experimental validation were raised and the authors have conducted more experiments in the rebuttal phase to strengthen it.

However, the AC/SAC note that doubts about the main claims remain even after the author response, so ultimately the recommendation is to reject the submission.